# The Greifswald Post COVID Rehabilitation Study and Research (PoCoRe)–Study Design, Characteristics and Evaluation Tools

**DOI:** 10.3390/jcm12020624

**Published:** 2023-01-12

**Authors:** Anke Steinmetz, Susanne Bahlmann, Corinna Bergelt, Barbara M. Bröker, Ralf Ewert, Stephan B. Felix, Agnes Flöel, Robert Fleischmann, Wolfgang Hoffmann, Silva Holtfreter, Matthias Nauck, Katja Riemann, Christian Scheer, Dana Stahl, Antje Vogelgesang, Uwe Völker, Ulrich Wiesmann, Johanna Klinger-König, René Walk, Hans J. Grabe, Stefan Gross, Kristin Lehnert, Jens Fielitz, Marcus Dörr

**Affiliations:** 1Physical and Rehabilitation Medicine, Department of Trauma, Reconstructive Surgery and Rehabilitation Medicine, University Medicine Greifswald, 17475 Greifswald, Germany; 2DZHK (German Center for Cardiovascular Research), University Medicine Greifswald, 17475 Greifswald, Germany; 3Department of Medical Psychology, University Medicine Greifswald, 17475 Greifswald, Germany; 4Institute of Immunology, University Medicine Greifswald, 17475 Greifswald, Germany; 5Department of Internal Medicine, Pulmonary Diseases, University Medicine Greifswald, 17475 Greifswald, Germany; 6Department of Internal Medicine B, University Medicine Greifswald, 17475 Greifswald, Germany; 7Department of Neurology, University Medicine Greifswald, 17475 Greifswald, Germany; 8Institute for Community Medicine, University Medicine Greifswald, 17475 Greifswald, Germany; 9Institute of Clinical Chemistry and Laboratory Medicine, University Medicine Greifswald, 17475 Greifswald, Germany; 10Department of Anaesthesiology and Intensive Care, University Medicine Greifswald, 17475 Greifswald, Germany; 11Interfaculty Institute of Genetics and Functional Genomics, University Medicine Greifswald, 17475 Greifswald, Germany; 12German Center for Neurodegenerative Diseases (DZNE), Site Rostock/Greifswald, 17475 Greifswald, Germany; 13Department of Psychiatry and Psychotherapy, University Medicine Greifswald, 17475 Greifswald, Germany

**Keywords:** post-COVID syndrome, non-hospitalized patients, rehabilitation, cardiopulmonary function, biobanking, psychological profile, 6-min-walk-test, cognitive function, after-care

## Abstract

(1) Background: COVID-19 is often associated with significant long-term symptoms and disability, i.e., the long/post-COVID syndrome (PCS). Even after presumably mild COVID-19 infections, an increasing number of patients seek medical help for these long-term sequelae, which can affect various organ systems. The pathogenesis of PCS is not yet understood. Therapy has so far been limited to symptomatic treatment. The Greifswald Post COVID Rehabilitation Study (PoCoRe) aims to follow and deeply phenotype outpatients with PCS in the long term, taking a holistic and comprehensive approach to the analysis of their symptoms, signs and biomarkers. (2) Methods: Post-COVID outpatients are screened for symptoms in different organ systems with a standardized medical history, clinical examination, various questionnaires as well as physical and cardiopulmonary function tests. In addition, biomaterials are collected for the analysis of immunomodulators, cytokines, chemokines, proteome patterns as well as specific (auto)antibodies. Patients are treated according to their individual needs, adhering to the current standard of care. PoCoRe’s overall aim is to optimize diagnostics and therapy in PCS patients.

## 1. Introduction

The highly infectious SARS-CoV-2, newly emerged in 2019, has caused a global pandemic with coronavirus disease 19 (COVID-19). This viral infection can cause prolonged symptoms and disabilities beyond the acute phase to a significant extent, referred to as Long and Post COVID Syndrome (PCS). In October 2021, WHO defined PCS as “occurs in individuals with a history of probable or confirmed SARS-CoV-2 infection, usually 3 months from the onset of COVID-19. Symptoms last for at least 2 months and cannot be explained by an alternative diagnosis.” [1]. The PCS Syndrome may affect COVID-19 patients after a severe acute course and hospitalization, as well as those with a presumably mild or asymptomatic acute disease without hospitalization.

A special challenge arises from the very heterogenic clinical presentation with a spectrum of symptoms that is similarly multifaceted as in the acute disease and that can affect several organ systems. So far, more than 200 different symptoms have been reported in connection with PCS [2]. Very often, the symptoms occur simultaneously in several organ systems and also lead to a substantial impairment of the quality of life [3,4]. The underlying pathogenesis is complex and not well understood. Among others, direct viral toxicity, (auto)immunological mechanisms, hypercoagulability and persistent inflammatory processes leading to endothelial lesions are discussed [5].

The incidence of PCS cannot yet be reliably estimated [6]. According to a recent Umbrella Review, PCS affects between 7.5% and 41% of adult COVID-19 patients without hospitalization and 38% of those who required hospitalization in the acute phase of the disease [7].

In February 2021, an interdisciplinary rehabilitation and therapy consultation was set up at the University Medicine Greifswald, Germany, for the medical care of non-hospitalized COVID-19 patients with long-term sequelae. The aim of this clinically motivated approach was to comprehensively diagnose affected patients and provide them with outpatient therapy and/or further therapies. In order to describe the clinical picture in detail and to accompany the consultation scientifically, an observational study was set up with a broad interdisciplinary orientation in all involved specialist departments. This study design is presented here.

The aim of the Greifswald PCS Rehabilitation Study and Research (PoCoRe) is to gain detailed bio-psycho-social (biological, clinical, psychological and social) insights into the course of PCS in patients who were not hospitalized during the acute illness. Using this multidisciplinary approach, different phenotypes of this heterogeneous syndrome will be identified and characterized by clinical as well as immunological or biomedical parameters. In the context of an outpatient rehabilitation concept, individualized therapy strategies will be evaluated and compared with the outcome of the current standard of usual care. 

## 2. Materials and Methods

### 2.1. Study Design

This is an ongoing monocentric prospective observational study with repeated measurement among patients experiencing PCS symptoms and attending the specialized PCS outpatient clinic of the Physical and Rehabilitative Medicine Centre of the University Medicine Greifswald after an acute COVID-19 infection. Within the framework of this setting, standardized basic diagnostics and resulting individual treatment, management and therapy will be carried out. Recruitment began in April 2021 and is ongoing. 

### 2.2. Eligibility Criteria

Patients with a proven PCR-positive SARS-CoV-2 infection and associated long-term symptoms were included. Patients with acute COVID-19 infection (persistent or recurrent) or acute infection less than 4 weeks ago, and those with insufficient language skills were excluded. A modular refusal to participate in individual parts of the examination by the patients was possible.

### 2.3. Ethics

The study was reviewed and approved by the Ethics Committee of the University Medicine Greifswald (No. BB 053-21) and was conducted in accordance with the current version of the Declaration of Helsinki [8]. The study was registered in the German Register for Clinical Studies (DRKS 00025007). Written informed consent was mandatory for study participation.

### 2.4. Study Population

Referrals to this outpatient clinic were made by general practitioners in the Greifswald region or by in-house medical officers, but patients could also be referred from outside the region. In addition, referrals were possible from patients who presented themselves in different special consultations of the University Medicine Greifswald or as inpatients for differential diagnosis or therapy of PCS symptoms.

### 2.5. Study Procedures

#### 2.5.1. Standardized Medical History Assessment and Medical Examination (PoCoRe Outpatient Clinic)

A standardized medical history and medical examination were carried out by standardized medical history and examination forms, containing information on the time of illness, symptoms of the acute infection, previous illnesses, current symptoms, social, family and work history, previous treatments, and vaccination status. In addition, an exploratory medical examination of the cardiovascular, musculoskeletal and neurological systems was carried out. The specific cardiopulmonary examination took place on a separate date at the examination center of the German Centre for Cardiovascular Research (DZHK) e.V., partner site Greifswald. All patients were offered an exploratory psychological consultation in the outpatient clinic of the Department of Medical Psychology of the University Medicine Greifswald. After the baseline examination, follow-ups with repeated assessments were carried out at 3 months, 6 months, 12 months, 18 months and 24 months after baseline (see Figure 1). 

#### 2.5.2. Vital Parameters and Physical Function Diagnostics

##### Vital Parameters

During the consultation, the following vital parameters were recorded: oxygen saturation, blood pressure, heart rate, and respiratory rate.

##### Six-Minute Walk Test

As part of functional performance diagnostics, a standardized 6-min walk test was carried out. In this test, the patients walk briskly for 6 min and the distance covered is measured using a running wheel. Afterward, the vital parameters oxygen saturation, heart rate and respiratory rate are measured again and the degree of exertion is recorded on the Borg exertion scale (range 6–20 points).

##### Ethyl Alcohol Threshold Test

To objectify disorders of olfactory perception, an olfactory test with an ethanol dilution series (10%, 25%, 50%, 70%, 96%) was performed according to a standardized and validated protocol [9].

##### Laboratory Measures and Biobanking

Various basic laboratory measurements were determined in routine patient care. These include blood count, inflammation markers, vitamin D as well as additional laboratory measurements at the physician’s discretion. Furthermore, additional biomaterials were collected and processed highly standardized in the core laboratory: 16 mL of whole blood for the isolation of peripheral blood mononuclear cells (PBMCs), as well as EDTA-plasma (4 mL), heparin plasma (4 mL) and serum (4 mL). The biomaterials were stored in our institutional biobank [10] at −80 °C in 2D codes cryovials for later analysis of immunomodulators, cytokines and chemokines, the plasma proteome as well as antibodies and beyond.

#### 2.5.3. Questionnaires

In order to reduce the time and psychomental burden of the questionnaires, they were handed out at different times or completed together during the consultation.

The following questionnaires were part of the regular diagnostics of the consultation, independent of study participation, and were completed by the patients on site before the consultation appointment:-MEDIAN Corona Recovery Score (GAD-7; PHQ-9, ITQ Part 1, critical life events),-Fatigue Assessment Scale-Veterans Rand 12-item Health SurveyThe COPD Assessment Test, the Post-COVID-19 Functional Status Scale as the well as the Montreal Cognitive Assessment are carried out by the medical staff during the consultation.

The following questionnaires will be handed out and filled in at the appointment of the cardiopulmonary examination in case of study participation:-Twenty-Item Toronto Alexithymia Scale-Childhood Trauma Screener-Generalized Resistance Resources (self-efficacy beliefs, stress experiences, self-esteem-short-scale, optimism-short-scale, social support-short scale, sense of coherence)

##### Median Corona Recovery Score

The MEDIAN Corona Recovery Score (MCRS) consists of three modules designed to comprehensively map the consequences of COVID-19. Module 1 (MCRS-S) includes physical functions such as oxygen saturation, respiratory rate, 6-min walk test and lung volume. Module 2 (MCRS-P) records psychological limitations. This contains the established and standardized questionnaires GAD-7 for Generalized Anxiety, PHQ-9 for Depression and ITQ Part 1 for Posttraumatic Stress Disorder (see below). Module 3 (MCRS-3) records so-called Corona-associated life events (critical life events). These comprise four events: L1. Corona-associated death of a key caregiver; L2. Corona-associated physical and/or social deprivation, e.g., through isolation, physical and social prohibitions on contact, temporary or permanent separation, etc.; L3. Corona-associated loss of autonomy (self-efficacy experience), e.g., through quarantine, curfews, bans on work, restricted movement, gaps in supply, monitoring, controls (health department, police, authorities), etc., and L4 Corona-associated loss of economic existence, e.g., loss of job, lack of remuneration, loss of assets, debt, bankruptcy.

The MCRS was developed in 2020 by a team from the MEDIAN Clinics with the aim of developing a new PROM tool for the rehabilitation and convalescence process after a corona disease [11]. The MCRS is used in many rehabilitation clinics throughout Germany; nevertheless, studies on reliability and validity are still pending.

##### Generalized Anxiety Disorder (GAD-7)

To assess symptoms of anxiety, we used the Generalized Anxiety Disorder (GAD-7; [12]). This screening questionnaire records anxiety symptoms and their severity over the past 2 weeks using a 4-point Likert scale (range 0–21 points). According to Spitzer a categorical evaluation defines “minimal” (0–4 points), “mild” (5–9 points), “moderate” (10–14 points) and “severe” [12].High validity and reliability have been reported [13,14].

##### Patient Health Questionnaire–Depression Module (PHQ-9)

Symptoms of depression were assessed with the Patient Health Questionnaire, which also refers to the twl preceding weeks. Answer options for the impairment by depression symptoms are possible on a 4-part Likert scale from “not at all” to “almost every day” (PHQ-9; [15]). A categorization according to Kroenke et al. is used: “minimal” (0–4), “mild” (5–9 points), “moderate” (10–14 points), “moderately severe” (15–19 points) and “severe” (20–27 points) (Kroenke et al. 2001). The PHQ-9 is highly valid and reliable [16,17,18].

##### International Trauma Questionnaire-Part I (Posttraumatic Stress Disorder)

The first part of the International Trauma Questionnaire (ITQ) was used for the assessment of post-traumatic stress disorder (PTSD) symptoms. The questionnaire comprises a total of nine questions from the symptom areas (1) re-experiencing in the here and now, (2) avoidance and (3) feeling of a current threat as well as the functional impairment of the same. The items are answered on a 5-point Likert scale from “not at all” (0) to “extremely” (4). The sum score ranges from 0 to 36 points, with higher scores indicating more PTSD symptoms. In order to be able to differentiate more precisely the degree of PTSD severity, the subscales are evaluated continuously, whereby values between 0 and 8 are possible for the individual subscales “re-experiencing”, “avoidance”, and “feeling of current threat”. Higher values indicate more PTSD symptoms. According to Cloitre et al. (2018), PTSD is plausible if at least 2 points are reached on each subscale and at least one of the three questions on functional impairment (items 7–9) is answered with “moderately”. The ITQ has been validated and measures reliable and clinically significant treatment-related changes in PTSD and complex PTSD [19]

##### COPD Assessment Test

The COPD Assessment Test (CAT) was developed to assess symptoms and impairment in patients with COPD [20]. It was evaluated for its applicability to patients with persistent pulmonary symptoms after COVID-19 infection and was found to be a useful assessment tool [21]. It captures a total of eight questions regarding the impact of pulmonary symptoms on well-being and daily life, which can be answered on a 6-point Likert scale ranging from 0 to 5. The sum score ranges between 0 and 40, with higher scores indicating more severe symptoms or impairment.

##### Post-COVID-19 Functional Status Scale

With the aim of having a simple instrument assessing functional limitations due to symptoms, anxiety, or pain, Klok et al. used a recently developed an ordinal scaled and validated score that measures functional limitations after venous thromboembolism (PVFS) as a template [22].

The Post COVID Functional Scale (PCFS) measures four levels of functional restrictions by asking about the limitations and restrictions resulting from the COVID-19 disease: Level 0 defines no functional impairment, i.e., no symptoms, pain, depression or anxiety, which have an impact on daily life. There exists a self-questionnaire and an interview version, the latter was used in this study. The opening question is: “How much are you currently affected in your everyday life by COVID-19?” using an algorithm of four questions, the functional status and the impairment within 4 degrees of severity can be determined within the framework of a structured interview.

Grade 1: Negligible functional impairment: All usual tasks/activities at home or at work can be performed with the same intensity despite some symptoms, pain, depression or anxiety.

Grade 2: Slight functional limitations: usual tasks/activities at home or at work are occasionally performed at a lower intensity or avoided due to symptoms, pain, depression or anxiety.

Grade 3: Moderate functional limitations: Usual duties/activities at home or at work have been structurally altered (reduced) due to symptoms, pain, depression or anxiety.

Grade 4: Severe functional limitations: Assistance is required with activities of daily living due to symptoms, pain, depression or anxiety: care and attention are required.

The PCFS has been used in PCS Research and has been evaluated in several studies, achieving. both good validity and reliability values [23,24,25,26]. The questionnaire and manual are freely available at https://osf.io/qgpdv/ (accessed on 11 December 2022) [27].

##### Fatigue Assessment Scale (FAS)

Fatigue symptoms and their degree of severity were assessed using the Fatigue Assessment Scale [28]. This questionnaire measures the severity of fatigue with five questions on physical and five questions on psychological fatigue symptoms using a 5-point Likert scale (sum score 10–50). To differentiate the severity of fatigue more comprehensibly, the following categories were used according to Michielsen et al. (2013): “no fatigue” (10–21), “fatigue” (22–34) and “severe fatigue” (35–50). The FAS is known to be highly reliable and valid [28].

##### Montreal Cognitive Assessment (MoCA)

To screen for cognitive deficits [29], the German version of the Montreal Cognitive Assessment (MoCA) was used. This contains a total of 30 questions and covers the cognitive domains of short-term memory, visuospatial abilities, language, abstract reasoning, executive functions, attention and concentration. Total scores between 0 and 30 can be achieved, with a score of 26 or above being considered normal. The test takes about 10 min to complete, and there are three different versions to avoid learning effects when the test is used several times. All patients who score below 26 points are offered a detailed neuropsychological assessment (for details see [30].

##### Informant Questionnaire on Cognitive Decline in the Elderly (IQCODE)

The IQCODE provides an assessment of cognitive changes by relatives and friends of the person. It was developed to assess cognitive decline and dementia in older persons and to provide information on pre-morbid cognitive levels. It has been proven to be a reliable and valid tool that is a good complement to established brief screenings [31]. The questionnaire contains a total of 16 questions that ask about memory aspects, acquisition of new memory content and access to existing knowledge compared to 10 years earlier (e.g., remembering birthdays, and learning to use new devices). These can be answered on a 5-point Likert scale between 1 (much improved), 3 (not much change) and 5 (much worse). Various cut-offs are used in the literature (range 3.3–3.6; [31]. The cut-off used here (3.4) according to Liem et al. [32] is calculated from the average of the individual questions.

##### Veterans Rand 12-Item Health Survey/Health-Related Quality of Life (HrQol)

The German-language Veterans Rand 12-item Health Survey (VR-12) was used to assess health-related quality of life (HrQoL). The VR-12 is a self-report questionnaire, that is as valid and reliable as the 12-item Short-Form Health Survey (SF-12) [33]. The 12 items of the VR-12 relate to a total of eight dimensions that capture physical, mental and social aspects of health. These include physical functioning, physical role functioning, pain, general health perception, vitality, social functioning, emotional role functioning and psychological well-being, which are to be answered using three- to six-point Likert scales. The standard version used here records the time span of the preceding 4 weeks. Two different scores can be calculated in the evaluation: the physical score and a psychological sum score. All twelve items are included in the calculation of both scores, whereby items with a physical reference have higher weights on the physical sum score and items with a reference to mental health have higher weights on the mental sum score. The score range of the questionnaire is between 0 and 100, with high sum scores representing good health. Furthermore, the VR-12′s responsiveness compares well with other established survey tools such as the EQ-5D or PROMIS [34].

##### Twenty-ITEM Toronto Alexithymia Scale (TAS-20)

Indications of alexithymia were assessed using the Twenty-Item Toronto Alexithymia Scale (TAS-20), which is based on a 5-point Likert scale and can range from 20–100 [35]. The higher the sum score, the more pronounced the alexithymia. The 66th percentile was used as the cut-off for differentiating between low and high alexithymia [36]. The TAS has been established and evaluated as a valid and reliable assessment tool [37].

##### Childhood Trauma Screener (CTS)

The Childhood Trauma Screener (CTS), an abbreviated form of the Childhood Trauma Questionnaire, was used to assess childhood abuse and neglect. It includes a total of five questions on physical, emotional and sexual abuse as well as physical and emotional neglect, which are asked on a 5-point Likert scale [38]. Cut-offs for the single CTS items are defined in Glasemer et al. [39]. According to Klinger-König et al. [40], dichotomous scores for abuse and neglect can be defined as none/mild vs. moderate/severe maltreatment reported on the three abuse and two neglect items, respectively.

##### Generalized Resistance Resources (GRRs)

Self-Efficacy Beliefs 

Self-efficacy was assessed with the Short Scale for Measuring General Self-efficacy Beliefs (ASKU; [41]), a self-report instrument that consists of three items rated on a 5-point scale ranging from 1 “does not apply at all” to 5 “fully applicable”. Higher scores indicate higher general self-efficacy.

Stress Experiences 

Current stress load was measured with eight items from the Stress and Coping Inventory [42], a self-report instrument that assesses selected stress-relevant areas in addition to the disease-specific burden: two items assessed stress caused by insecurities (financial problems, insecurities with regard to place of residence), three items assessed overload (e.g., job strain, demands by family/friends, own demands and expectations), and three items assessed loss experiences (job loss, loss of family members/friends, loss of or separation from partner). Each item was rated on a 7-point scale ranging from 1 “did not happen/no burden” to 7 “very strong burden”.

Self-Esteem-Short-Scale 

Self-esteem was measured with the Single-Item Self-Esteem Scale [43], containing a self-reported item “I have high self-esteem”. It is rated on a 7-point Likert scale from 1 (not very true of me) to 7 (very true of me). A high score indicates a high level of self-esteem.

Optimism-Short-Scale 

Optimism was measured with the German Version of the Life Orientation Test (LOT-R, [44], a self-report instrument that consists of six items assessing optimistic beliefs (e.g., “In certain times, I usually expect the best.”) and pessimistic expectations (e.g., “If something can go wrong for me, it will.”). which are assessed on a five-point scale from 1 (exactly true) to 5 (not at all true). A high sum score represents a high level of dispositional optimism.

Social Support-Short Scale 

Social support was measured with the Short Form of the Social Support Scale (F-SozU K-14; [45,46], a self-report instrument consisting of 14 items. Each item is rated on a 5-point scale from 1 “exactly true” to 5 “not at all true”. Item ratings are to be inverted and then summarized into a global social support score. A high sum score reflects a high level of social support.

Sense of Coherence 

SOC was measured with the Leipzig Short Form of Antonovsky’s Sense of Coherence Scale (SOC-L9) [47], a self-report instrument that incorporates nine items assessing the extent of comprehensibility, manageability and meaningfulness on a 7-point scale. An item example is: “How often do you have feelings that there’s little meaning in the things you do in your daily life? Often vs. seldom or never”. A high sum score represents a strong sense of coherence.

#### 2.5.4. Cardiopulmonary Examination Modules

##### Anthropometric Parameters

Body weight, body mass index (BMI), hip circumference and waist-to-hip ratio were measured according to standard operating procedures.

##### Arm, Thigh and Lower Leg Measurement

All measurements were performed on the relaxed patient in a supine position, values were collected from three measurements with a measuring tape. Measurement of the mid-upper arm circumference was performed in the middle between the acromion and olecranon on the 90° angled arm. The mid and two third of the upper thigh circumferences were measured in the middle or two third distance between the anterior iliac spine and the upper margin of the patella. Mid-lower leg circumference was measured in the middle between the lower margin of the patella and lateral ankle.

##### Ultrasound Measurement of Mid and Two Third of Upper Thigh

Ultrasound measurements were performed with a Vivid E9 (GE Healthcare, Solingen, Germany) ultrasound device; the probe (9L) was placed on the middle or two third distance between the anterior iliac spine and upper margin of the patella; the femoral bone was centered in the middle of the frame; the muscle was compressed until no distance change was visible and a still frame was saved. Distance measures were performed offline according to the leading-edge method.

##### Bioelectrical Impedance Analysis (BIA)

Body composition analysis was performed with a Nutriguard-MS device and NutriPlus analysis software (Data Input GmbH, Pöcking, Germany). Skin electrodes were placed on the patient’s hand and foot in a supine position.

##### Handgrip Strength

The handgrip test was performed with the Jamar Plus Digital Dynamometer^®^ (Patterson Medical, Warrenville, IL, USA) according to a standardized protocol. It started with the measurement of the hand grip strength of the right hand and continues with the measurement of the left hand. A total of three runs were completed. The pause between measurements of the same hand was at least 15 s. The dominant hand was tested and documented.

##### Pulmonary Function

Lung function testing was performed with a MasterScreen Body plethysmography system (Jaeger, CareFusion, Höchberg, Germany) and SentrySuite analysis Software (Vyaire Medical, Höchberg, Germany). For this purpose, spirometry, body plethysmography including diffusion analysis as well as the measurement of muscular respiratory drive and respiratory muscle strength were performed.

##### Echocardiography

Two-dimensional transthoracic echocardiography was performed as a non-invasive gold standard to determine the function and morphology of the heart and to assess the function of the heart valves according to the DZHK echocardiography standard operating procedure (SOP) [48]. A Vivid E9 ultrasound device (GE Healthcare, Solingen, Germany) was used and measurement of the parameters in the recorded standard views were performed as posthoc reading.

##### Pulse Wave Analysis and of Vascular Stiffness

A non-invasive cuff-based method (Vascular Explorer, Enverdis GmbH, Germany) was used for pulse wave analysis and determination of arterial stiffness parameters. All measurements were performed under standardized conditions after a resting period in a lying position on the right upper arm. The device software calculated the following arterial stiffness parameters: pulse wave velocity (PWV) brachial-ankle (PWVba), aortic/central PWV (PWVao), carotid-femoral PWV (PWVcf), augmentation index (AiX) brachial (AiX_br), aortic/central AiX (AiX_ao), heart rate corrected aortic/central AiX (AiX@75).

In addition, brachial blood pressure values were determined and aortic/central blood pressure values were calculated by the device software.

#### 2.5.5. Data Collection, Management and Analysis

Medical history data, questionnaire and test results collected by paper and pencil were entered into an Excel-based datasheet under the corresponding study pseudonym. The examination results of the appointment at the cardiology examination center were documented digitally as well. Subsequently, both data sources were merged and processed for statistical analysis.

The patients were given follow-up appointments by telephone, and if they drop out of the study, they were asked for their reasons for dropping out.

## 3. Discussion

The aim of this interdisciplinary study is to investigate PCS with a comprehensive research approach, which transcends the traditional organ-centered perspective. Interdisciplinary networking and translational approaches are to be used to deepen our understanding of PCS, improve diagnostics and develop targeted therapy concepts. Guided by basic diagnostics, specific functional analyses are carried out. Within the framework of translational approaches, the associations of immunological biomarkers (specific (auto)antibodies, immunomodulators, cytokines and chemokines, plasma proteome profiles, etc.) with symptoms, organ manifestations and therapeutic effects will be investigated. Patients presenting at a PCS rehabilitation consultation receive interdisciplinary medical support to investigate mechanisms, phenotypes and the PCS disease course. Within our framework, the examinations and tests described are the basis for standardized comparisons; nonetheless, therapies are recommended and carried out to meet the individual patient’s needs. They are based on the current state of knowledge and specifically address the individual symptoms, performance limitations and organ manifestations. The overall aim of this broad-based study is to optimize diagnostics and therapeutic options for patients affected by PCS.

Existing cohort studies that aim to assess long-term symptoms after COVID-19 disease partly include much higher numbers of cases, but are mostly based either on self-report only or on the use of electronic health records, often follow the participants for no longer than a year and do not necessarily have as broad a multidisciplinary approach as the one used in our study [49,50,51,52,53,54]. Population-based approaches make an important contribution to the investigation of (long-term) symptoms associated with PCS. For example, recently published results of a questionnaire-based survey of 11,710 PCS patients showed that six to 12 months after acute SARS-CoV-2 infection, even in young and middle-aged adults after a mild infection, there is a significant burden of self-reported post-acute symptom groups and possible sequelae, particularly fatigue and neurocognitive impairment [54]. The advantage of the large samples of such studies is countered by the lack of complementary medical examinations, such as those collected in our study, which could allow further considerations on the pathogenesis of symptom persistence. These limitations also apply to population-based approaches using large datasets of healthcare data [53]. Other existing studies have also collected medical data but usually focused on a specific research area. For example, a recent study from Heidelberg, Germany, reports the 12-month follow-up data of 96 patients with PCS [55]. Only 12 (22.9%) of the participants were completely symptom-free at this time, while all others (n = 84) still reported health problems. Neurocognitive impairment was the most commonly reported symptom, which was also associated with elevated ANA titers, suggesting autoimmune processes are involved in the etiology. Another example is the recently published neurological follow-up data of the first 100 patients of a special neurological post-COVID consultation at Charité Berlin, Germany, showing that neurological sequelae often persist beyond 3 months even after mild acute SARS-CoV-2 infections. [45].

In addition to the above-mentioned limitations of current follow-up studies of PCS patients, symptomatic therapies administered during the follow-up period are usually not systematically recorded and evaluated; this will be included in the current study design.

### 3.1. Limitations and Biases

The complex and comprehensive study design poses serious challenges to the motivation and commitment of the participants. On the one hand, the high number of questionnaires and the resulting psychomental stress on the patients should not be underestimated. Nevertheless, a standardized survey of various symptom complexes is also indispensable for routine diagnostics outside the study, thus a large part of the questionnaires is regularly collected from all patients during the consultation. However, in order to reduce fatigue effects, the questionnaires are given out at different times. Some of the questionnaires are completed by the patients during the waiting time before the appointment, and others together with the medical staff during the consultation. Additional study questionnaires are given out at the second appointment of the cardiopulmonary examination.

Another challenge is to achieve the highest possible study adherence and to minimize the dropout rate, which is often high in such studies. In order to control the drop-out, during the course the study participants are contacted by telephone shortly before the next scheduled visit (possibly several times); in case of scheduling problems, the appointment will be rescheduled in agreement with the subjects. In the case of a drop-out, the reasons (e.g., no more complaints, no time/interest) are recorded.

More patients with severe courses may attend a specialized outpatient clinic. The higher utilization of patients with secondary illness gains or pre-existing severe underlying mental illnesses cannot be ruled out either. Therefore, selection bias is a potential confounding factor. We, therefore, plan to record factors that lead to participation or non-participation in our study in order to be able to take these into account in the analyses.

Furthermore, generalizability may be limited to the observational study design and its single-center and nonrandomized character.

### 3.2. Perspective

PCS usually involves several organ systems, which makes interdisciplinary diagnosis and therapy necessary. Intensive research has been carried out in the individual disciplines to clarify the pathomechanisms of PCS. Nevertheless, there is an urge to find a holistic pathophysiology to address the individual needs of patients calling for a new paradigm explaining PCS. Thus, Saunders et al. [56] suggest different biological, psychological as well as social and environmental factors contributing to the development of PCS.

The aim of this interdisciplinary research approach is to deepen the understanding of PCS through a cross-organ approach and networking of research results. In our view, this strategy is an important starting point for developing causal therapy approaches for PCS and improving rehabilitation. We are very confident that this approach will significantly support the process of elucidating the complex pathomechanisms of PCS.

## 4. Conclusions

PoCoRe is an important study that analyzes PCS under a broad comprehensive view in order to optimize diagnostics and therapy options.

## Figures and Tables

**Figure 1 jcm-12-00624-f001:**
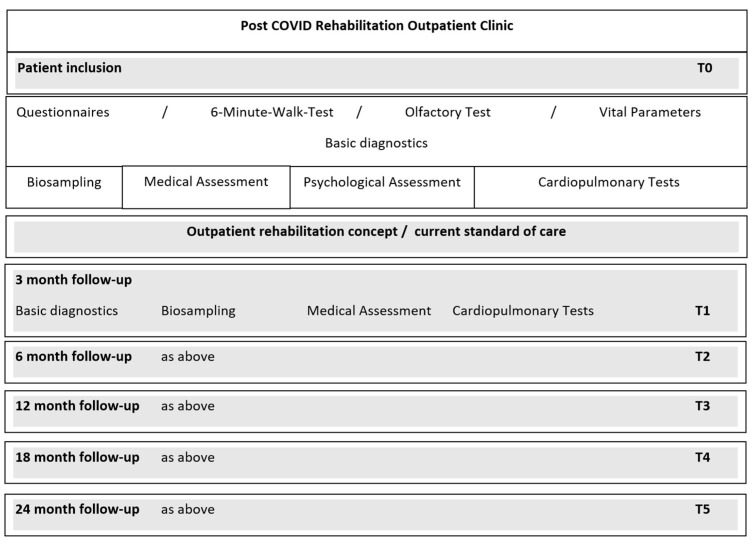
Study procedures.

## Data Availability

Data of completed analyses will be available from the corresponding author upon reasonable request.

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
