# Peer review of "The Greifswald Post COVID Rehabilitation Study and Research (PoCoRe)–Study Design, Characteristics and Evaluation Tools"

_jcm, 2023, doi:10.3390/jcm12020624_

Round 1
Reviewer 1 Report
Dear authors,
I appreciate your intention to perform such an extensive research, and I attached my comments bellow.
Best regards

Author Response
Point-by-point response to the reviewer’s comments
We would like to thank both reviewers for their very helpful comments and the effort they put into reviewing our manuscript. We believe that we have been able to improve the manuscript by revising it based on these suggestions.
Reviewer 1:
“I read with great interest the proposal for your study and I greatly appreciate the amount of data you want to collect, but also the huge amount of work it entails.
The sub-acute and long COVID-19 syndromes are newly emerged entities that have been debated quite a lot in the last period of time in the specialized medical literature, but which still present many unknown or insufficiently clarified aspects. Several scientific articles have already been published on this topic, some of them focusing on the cardiovascular and neuro- psychological consequences of the post-COVID-19 syndrome, and of the impact on patients' quality of life, as, for example, in the article of Giurgi-Oncu, C.; Tudoran, C.; Pop, G.N.; Bredicean, C.; Pescariu, S.A.; Giurgiuca, A.; Tudoran, M. Cardiovascular Abnormalities and Mental Health Difficulties Result in a Reduced Quality of Life in the Post-Acute COVID-19 Syndrome. Brain Sci. 2021,11,1456.
I appreciate the intention of the proposed study to follow the patients for 24 months through so many procedures, but I think that the participants' adherence to the protocol will represent a problem.”
Response:
Thank you very much for your comment and the valuable reference you mentioned. We have revised the introduction, as suggested, in order to emphasize more clearly the wide range of symptoms that occur in the context of the Post/Long Covid Syndrome (see changes in tracking modus). We have also tried to point out the interdisciplinary nature of our project, which could be instrumental in generating new insights into this syndrome. In this context, we have added both the recommended (Giurgi-Oncu et al. 2021) and another reference (Petersen et al. 2022) to better characterize the multidisciplinary appearance of Post COVID symptoms 8 (line 74).
We understand the concerns about participant adherence and are aware that there is a risk of high drop-out rates. Therefore, we make a great effort to maintain close contact with the participants. For example, we contact them by phone before the respective follow-up appointments. We also ask about the reasons for early drop-out from study participation. During the course of recruitment and follow-up so far, we have made the experience that these measures contribute to a high participation rate. We have added a corresponding section on the "challenges" of the study design in the discussion to take this important aspect into account (lines 507-522).
References:
Giurgi-Oncu, C.; Tudoran, C.; Pop, G.N.; Bredicean, C.; Pescariu, S.A.; Giurgiuca, A.; Tudoran, M. Cardiovascular Abnormalities and Mental Health Difficulties Result in a Reduced Quality of Life in the Post-Acute COVID-19 Syn-drome. Brain Sci. 2021, 11.
Petersen, E.L.; Goßling, A.; Adam, G.; Aepfelbacher, M.; Behrendt, C.-A.; Cavus, E.; Cheng, B.; Fischer, N.; Gallinat, J.; Kühn, S.; et al. Multi-organ assessment in mainly non-hospitalized individuals after SARS-CoV-2 infection: The Hamburg City Health Study COVID programme. Eur. Heart J. 2022, 43, 1124–1137.
Reviewer 2 Report
A protocol for a very ambitious cohort study. The research question is clear and the design broadly appropriate. My one major concern is that the researchers plan to ask people with long covid (who are known to get tired - indeed fatigue and fatiguability are the commonest symptoms) to complete (I think) 21 different questionnaires six times each over an 18-month period. I have major ethical concerns about this (long covid patients develop post-exertional symptom exacerbation after physical OR cognitive stress), and also I am unconvinced that the responses to this number of questionnaires will be valid (people will get bored or tired and not fill them out properly). Also, the MEDIAN corona recovery score has not been validated. Why not use the PROM that has (Manoj Sivan's C19-YRS which is the one used by WHO and is becoming the standard PROM for long covid)? These are my major concerns about an otherwise well-designed study.
Author Response
Point-by-point response to the reviewer’s comments
We would like to thank both reviewers for their very helpful comments and the effort they put into reviewing our manuscript. We believe that we have been able to improve the manuscript by revising it based on these suggestions.
Reviewer 2:
“A protocol for a very ambitious cohort study. The research question is clear and the design broadly appropriate. My one major concern is that the researchers plan to ask people with long covid (who are known to get tired - indeed fatigue and fatiguability are the commonest symptoms) to complete (I think) 21 different questionnaires six times each over an 18-month period. I have major ethical concerns about this (long covid patients develop post-exertional symptom exacerbation after physical OR cognitive stress), and also I am unconvinced that the responses to this number of questionnaires will be valid (people will get bored or tired and not fill them out properly). Also, the MEDIAN corona recovery score has not been validated. Why not use the PROM that has (Manoj Sivan's C19-YRS which is the one used by WHO and is becoming the standard PROM for long covid)? These are my major concerns about an otherwise well-designed study.”
Response:
Thank you very much for your valuable comments and we can well appreciate the concerns about the burden on the subjects. To counter this risk, we decided to split the distribution of the questionnaires during the study planning. Thus, some of the questionnaires are filled out by the participants in the waiting room before the appointment, others are completed together with the medical staff during the consultation. Only questionnaires that are part of the regular consultation and are filled out by patients independently of their participation in the study, are collected during the first consultation appointment. The questionnaires, which are exclusively relevant to the study, are given to the patients at the second examination appointment as part of the cardiopulmonary diagnostics. Our experience of almost two years in the context of this consultation shows that this has fortunately not led to the occurrence of post-exertional symptoms in our patients by now. We have clarified this procedure in the methods section and added a paragraph in the discussion in which we address this potential issues (lines 507-522).
When planning the study in February 2021, we decided to use the MCRS (MEDIAN corona recovery score) as a PROM because it contains established and validated assessment tools for psychological diagnostics, which have already been used extensively in Post COVID research (GAD-7, PHQ-9 e.g. in Petersen et al. 2022, De Luca et al. 2022, O' Mahony et al. 2022). Nevertheless, thank you for the valuable reference to the C19-YRS, which unfortunately we were not aware of at the time of planning the study and which was apparently only validated at that time.
References:
De Luca R, Bonanno M, Calabrò RS. Psychological and Cognitive Effects of Long COVID: A Narrative Review Focusing on the Assessment and Rehabilitative Approach. J Clin Med. 2022;11(21):6554.
O' Mahony L, Buwalda T, Blair M, et al. Impact of Long COVID on health and quality of life. HRB Open Res. 2022;5:31.
Petersen, E.L.; Goßling, A.; Adam, G.; Aepfelbacher, M.; Behrendt, C.-A.; Cavus, E.; Cheng, B.; Fischer, N.; Gallinat, J.; Kühn, S.; et al. Multi-organ assessment in mainly non-hospitalized individuals after SARS-CoV-2 infection: The Hamburg City Health Study COVID programme. Eur. Heart J. 2022, 43, 1124–1137.
Round 2
Reviewer 2 Report
They have responded carefully and appropriately to my initial concerns. Happy for this to be published and I look forward to seeing the results.
Author Response
Thank you very much for your positive feedback. We would like to thank you once again for taking the time to read and review our manuscript.